# CO_2_-Assisted Sugar Cane Gasification Using Transition Metal Catalysis: An Impact of Metal Loading on the Catalytic Behavior

**DOI:** 10.3390/ma16165662

**Published:** 2023-08-17

**Authors:** Daria A. Beldova, Artem A. Medvedev, Alexander L. Kustov, Mikhail Yu. Mashkin, Vladislav Yu. Kirsanov, Irina V. Vysotskaya, Pavel V. Sokolovskiy, Leonid M. Kustov

**Affiliations:** 1Chemistry Department, Moscow State University, 119992 Moscow, Russia; dashabelk@yandex.ru (D.A.B.); artom.medvedev@yandex.ru (A.A.M.); mikhail.y.mashkin@gmail.com (M.Y.M.); levap90@list.ru (P.V.S.); lmkustov@mail.ru (L.M.K.); 2N. D. Zelinsky Institute of Organic Chemistry RAS, 119991 Moscow, Russia; 3VNIIneft JSC, Scientific and Technological Center, EOR Department, 127422 Moscow, Russia; 4Laboratory of Nanochemistry and Ecology, Institute of Ecotechnologies, National University of Science and Technology MISIS, 119071 Moscow, Russia; 5N.V. Sklifosovskiy Institute of Clinical Medicine, I.M. Sechenov First Moscow State Medical University, 119991 Moscow, Russia; kirsanov_v_yu@staff.sechenov.ru (V.Y.K.); vysotskaya_i_v@staff.sechenov.ru (I.V.V.)

**Keywords:** sugar cane bagasse, CO_2_ utilization, carbon wastes utilization, iron catalysis, cobalt catalysis, nickel catalysis

## Abstract

To meet the increasing needs of fuels, especially non-fossil fuels, the production of “bio-oil” is proposed and many efforts have been undertaken to find effective ways to transform bio-wastes into valuable substances to obtain the fuels and simultaneously reduce carbon wastes, including CO_2_. This work is devoted to the gasification of sugar cane bagasse to produce CO in the process assisted by CO_2_. The metals were varied (Fe, Co, or Ni), along with their amounts, in order to find the optimal catalyst composition. The materials were investigated by scanning electron microscopy (SEM), energy-dispersive X-ray spectroscopy (EDX), transmission electron microscopy (TEM), X-ray diffraction (XRD), and electron diffraction, and were tested in the process of CO_2_-assisted gasification. The catalysts based on Co and Ni demonstrate the best activity among the investigated systems: the conversion of CO_2_ reached 88% at ~800 °C (vs. 20% for the pure sugarcane bagasse). These samples contain metallic Co or Ni, while Fe is in oxide form.

## 1. Introduction

CO_2_ emissions are one of the most important problems of recent years due to the extremely large CO_2_ production and its role in the greenhouse effect. There are several reactions available to transform CO_2_ into other chemical forms of carbon, such as syngas, methanol, dimethyl ether, urea, dimethyl carbonate, polyurethane, etc. [1,2]. An application of such reactions seems to be a possible way to partially decrease an atmospheric emission, but it takes a great deal of energy to perform high temperature processes. This results from the chemical stability of the CO_2_ molecule and it is necessary to use highly reactive agents such as hydrogen or apply heating to overcome its inertness. Moreover, CO_2_ is a higher oxidation state of carbon atom and can be applied as a mild oxidant in a series of the processes such as carbon materials gasification (Equation (1)), alkanes dehydrogenation, etc. Its application in this role was widely described in the literature [2,3,4,5]. Thus, it seems reasonable to try to apply CO_2_ as an oxidant to the process of CO_2_-assisted gasification of carbon wastes [6].
C_(s)_ + CO_2(g)_ = 2CO_(g)_(1)

Since the demand on energy and therefore on the fuels has increased significantly in recent decades and further growth is predicted, the necessity of searching for effective and ecologically friendly ways to meet needs with non-fossil fuels is obvious [7,8]. There are different approaches to their utilisation, for example using them as a raw materials (but it takes using the noble metal catalysis) [9,10]. Despite a large amount of researches in this field, especially on biomass pyrolysis and on biomass conversion in supercritical water [11,12], the complex nature of a biomass of any kind [13] leads to difficulties in its conversion into valuable products (instead of burning it to produce heat) and the effective catalytic system should be found. The amounts of biomass produced annually is impressive (in the case of sugar cane it reaches 1.6 billion tons annually), and the amount of this biomass residue is described as about 280 million tons annually [14]. Steam gasification and pyrolysis are the main non-biotechnological ways used in the utilisation of the sugarcane bagasse and other residues such as leaves, etc. [14]. Bio-oils produced in these processes can also be improved and further used as noble metal catalysis [15].

Sugar cane bagasse (further denoted as SCB) is a large scale by-product of the sugar industry for which annual production reaches 513 million tons [16]. The first attempts to model the sugarcane gasification mills revealed the potential benefits of the realisation of such a technology [17]. Parametric energy and exergy (first and second low analyses) studies of sugar cane bagasse gasification resulted in the following conclusion: the less the amount of water in biomass, the less the exergy losses in the reactor, while other parameters do not influence the exergy efficiency of the process as much [17]. Moreover, the moisture content can be up to 50%, so the drying pre-treatment should be conducted [18]. The pellet size can affect the gasification rate but not the pyrolysis rate, as was found recently [19]. The different ways to implement such a technology are proposed and modelled with Aspen Plus software, and it was found that the most impactful parameters were the temperature and the ratio of the gasifying agent (it was steam in the reported work) to sugar cane bagasse [20]. Generalised biomass (here sugar cane bagasse) transformation is proposed to be described as follows. The first step is biomass pyrolysis, resulting in char, tar, and volatile compounds; char is further subject to gasification while tar undergoes cracking and reforming, along with volatiles [21,22,23].

The main efforts in catalytic processes development in the carbon wastes utilisation are focused on the following groups: alkali metals (for example, [24,25,26,27,28,29]), alkali–earth metals [30,31,32], and transition metals [33,34,35,36,37,38,39,40,41,42,43]. Different natural and synthetic materials have been tested as carriers for the catalyst for carbon waste utilisation, for example, dolomite [44], olivine [45,46], or MgO [47], etc., but the main research is focused on the systems with metals directly deposited on the gasified (or pyrolyzed) material such as rice husk, etc., for example [48,49,50,51]. SiO_2_ and Al_2_O_3_ can affect the catalytic behaviour as well [52]. The different combinations of transition metals are also presented in the recent studies, for example Fe-Mo [53]. It was reported that surface morphology and phase composition and position on a surface play a role in the wood gasification process [54]. Both the material parameters and gasifier conditions affect the syngas yield and composition [55]. Interesting results have been reported on co-manganese spinel in low pressure CO_2_ hydrogenation: the catalyst is active and selective to methane [56]. Despite all the efforts conducted, further investigation and industrial adaptation of these processes still needs to be performed.

The main novelty of this work is related to revealing the correlation between the nature of the metal deposited on sugarcane bagasse, the resulting structural properties, and the catalytic activity of the materials.

This work is aimed at the preparation and investigation of the catalytic systems based on sugar cane bagasse and the transition metal (Fe, Co, or Ni) to reveal the dependence of the catalytic behaviour on the metal nature in the catalytic CO_2_-assisted gasification.

## 2. Materials and Methods

The following reagents were used in this work: Fe(NO_3_)_3_·9H_2_O (99%), Co(NO_3_)_2_·6H_2_O (99%), and Ni(NO_3_)_2_·6H_2_O (98%) from Acros Organics, sugar cane bagasse, and bi-distilled water. All the reagents were used as purchased without further purification. The sugar cane bagasse was a waste of the technological process of sugar cane juice production. The material was donated in kind by the Institute of Environmental Technology, Vietnam Academy of Science and Technology.

All the materials were examined by scanning electron microscopy with energy-dispersive X-ray spectroscopy using a Leo Supra 50VP (Carl Zeiss, Gottingen, Germany) scanning electron microscope under a low vacuum in a nitrogen atmosphere. EDX data were collected using an energy-dispersive spectrometer INCA Energy (Oxford Instruments, X-Max-80, Abingdon, UK).

Transmission electron microscopy studies of the materials were performed using a transmission electron microscope JEM-2100 JEOL (Tokyo, Japan). Electron diffraction patterns were processed with the software package [57].

Powder X-Ray diffraction patterns were collected with a STOE STADI P transmission diffractometer (Chicago, IL, USA) using CuKα1 radiation (λ = 1.54056 Å) monochromatized with a curved germanium (111) monochromator. The samples were examined in the region 2θ = 10–60°, with a step of 0.01° and a 10 s counting time per point. Before the examination, the samples were heated at 600 °C for one hour in a flow of CO_2_ of 30 mL per minute.

The evaluation of the activities of the resulting materials in the gasification process was performed using a quartz flow-type reactor with the internal diameter of 8 mm under the CO_2_ pressure of 1 atm. The temperature ramp was 10 °C per minute, the temperature range was 100–850 °C, and the total flow rate of CO_2_ was 30 mL per minute. A Bronkhorst EL-FLOW SELECT F-111B (Leonhardsbuch, Germany) gas flow controller was used to control the gas flow rate. The material loading was 1 g. Although natural industrial material particles can be different sizes [41], here the particle size was chosen to be 0.5–1 mm. The reaction gas products were analysed using a Chromatek Crystal 5000 (Yoshkar-Ola, Russia) gas chromatograph with thermal conductivity detectors, M ss316 3 m × 2 mm columns, Hayesep Q 80/100 mesh, and CaA molecular sieves.

The conversion (*X*) of carbon dioxide during the tests was calculated by the following formula (Equation (2)).
(2)$$X_{{CO}_{2}}=\frac{F_{{CO}_{2}}^{in}-F_{{CO}_{2}}^{out}}{F_{{CO}_{2}}^{in}}$$

### Synthetic Procedure

All the samples were prepared as follows: the nitrate of Fe, Co, or Ni was completely dissolved in the appropriate amount of water to obtain the volume of the solution equal to the incipient wetness capacity measured previously. The amount of sugar cane bagasse (5 g) was impregnated with the appropriate salt solution amount to obtain the desired metal content (1, 3, or 5 wt. %). The samples were dried at 50 °C overnight in an oven. Therefore, the series of 10 samples was prepared and further designated as *nM*/SCB where *n* was 1, 3, or 5 wt. % of metal loading and *M* was Fe, Co, or Ni.

## 3. Results and Discussion

### 3.1. Scanning Electron Microscopy

The results of SEM and EDX characterisation are presented in Figure 1. All the samples have a structure with channels that typically present in the plants. As can be clearly seen, the samples differ from each other in both homogeneity of the metal distribution over the surface and the morphology. The samples with 1 wt. % metal loading showed some correlation between the position in the grain of the material and the metal concentration. As, for example, can be seen in the picture for 1Ni/SCB, the outer surface of the particle is covered with Ni compounds in a higher concentration than its inner surface. In the case of 1Fe/SCB and 5Fe/SCB samples, we can see that some channels have different transparencies towards the electron beam. Nevertheless, the observation that the surface layers have a larger concentration seems to be right, and, in the case of iron loaded with 1 and 3 wt. % samples, the depth of these layers is up to 50 mm irrespective of the percentage. But when SCB is loaded with 5 wt. % of the metal, the distribution of the metals seems to be more uniform. 

Therefore, the main conclusion from this study is that the Ni-loaded samples demonstrate the least uniformly distributed metal oxide particles on the surface. The distribution of particles on the surface of the Co-loaded sample is more uniform and the most uniform one was found for the iron-loaded catalyst.

### 3.2. Transmission Electron Microscopy

TEM images (Figure 2) show a clear difference between the samples: in the case of Co- and Ni-loaded systems, relatively large particles can be seen, while only small particles are revealed in the case of the iron-loaded sample. It is of note that in the case of the Ni-loaded sample, the fraction of small observable particles is much larger than in the case of the Co-loaded catalyst. The particle size distributions show that in each case the maximum of the distribution tends to be observed in the region of small particles and increases in the following order: Fe < Ni < Co. The particle size distributions show a unimodal shape with an average size for samples under investigation (5% mass metal loading). 

TEM images (Figure 2) show clear difference between the samples: in the case of Co- and Ni-loaded systems the relatively large particles can be seen, while in the case of the iron-loaded system only the small particles were revealed. It is noteworthy that in the case of the Ni-loaded sample, the fraction of small observable particles is much larger than in the case of Co-loaded catalyst. The particle size is distributed monomodally, and average sizes for the samples under investigation (5% mass metal loading) were of about 9.7, 53.3, and 23.5 nm for the samples loaded with Fe, Co, and Ni, respectively. The same dependence on metal particle size was observed in the previous works where Ni formed much smaller particles than cobalt [58].

### 3.3. XRD and Electron Diffraction Patterns

XRD patterns for the series of the samples with 5% mass metal loading before the catalytic tests (after pre-heating at 300 °C; in a CO_2_ flow) (Figure 3) show that all the samples demonstrate a kind of a halo in the region of about 2θ = 20–25°, probably corresponding to amorphous silica that is naturally present in the plants. Also, all the samples contain silica as a cristobalite phase (the ICDD card number [46-1242]). In the case of Co- and Ni-loaded materials, the metallic phases of Co and Ni can be seen. This fact can result from the reductive ability of the sugarcane bagasse towards metal ions transformation into metal particles. It can be observed especially clearly for the sample Ni/SCB.

The electron diffraction patterns were also obtained during TEM examination (Figure 4). The pattern for the sample Co/SCB corresponds to the metallic Co phase according to the ICDD database, card number [15-806]. These results are consistent with the results of XRD: the metallic phases can also be seen in XRD patterns as well. The diffraction patterns for the sample Fe/SCB contain weak signals that are too small and cannot be used as firm evidence. Nevertheless, the only observed dots correspond to the interplanar distance of 2.02 A, which can be possibly attributed to the phase of metallic iron (the reference d-spacing is 2.027 A [6-696]). Despite this, we cannot prove it because any ex situ methods here imply contact with oxygen, while small iron particles might be unstable under these conditions.

For the series of the samples after the catalytic tests in CO_2_-assisted gasification, XRD investigation (Figure 5) shows similar patterns to those obtained before the catalytic tests, but some differences are clearly seen. In the case of the sample Co/SCB, the new phase appeared—CoO (the ICDD card number [43-1004]) and the metallic Co phase are still present in the sample. In the case of the Ni-loaded material, the average crystallite size calculated by the Sherrer equation slightly increased and reached ~35 nm (vs. ~30 nm for the sample after pre-heating). Electron diffraction patterns (Figure 6) showed the same phases for the samples with Ni and Co, but, in the case of the iron-containing sample, the phase of maghemite Fe_2_O_3_ (the ICDD card number [25-1402]) was observed. Nevertheless, since small iron particles are pyrophoric, the formation of oxide can be the result of oxidation of the metal nanoparticles reduced under reaction conditions, but this is only a tentative hypothesis and was not proved here. The formation of metallic phases is in accordance with the literature data, for example, [51]. Pure SCB was additionally tested in gasification under the same conditions, and it was found that, presumably, the phase of calcium phosphate Ca_2_P_2_O_7_ (the ICDD card number [9-346]) is present in the sample. This result does not contradict the results of EDX where the following elements were found—K, Ca, Mg, Si, and P. All the elements naturally occur in the sample, as was also reported in the literature [20,59].

### 3.4. Catalytic Tests

The results of the catalytic tests are shown in Figure 7. The samples clearly differ from each other: both Co- and Ni-loaded samples work much better than Fe-loaded one and the blank one. Moreover, the catalytic behaviour of the samples Co/SCB and Ni/SCB is nearly the same over the entire investigated temperature range. Non-zero conversion for the blank sample might result from the biological materials not being loaded with metals that still contain some of them naturally (for example, K, Ca, Mg, Fe), which have their own catalytic activities in this process. Nevertheless, the activity of metal-loaded samples is much higher than that of the samples without loaded metals, possibly because of a kind of synergism between the transition metal and alkali (or alkali earth) metals.

In comparison, the Co- and Ni-loaded materials demonstrate a relatively close catalytic behaviour: it may indicate that three wt. % metal loading is the optimal value. Nevertheless, the five wt. %-loaded samples do not work less well than the three wt. %-loaded materials.

There seems to be a kind of correlation between the formation of relatively stable metal nanoparticles and high catalytic activity in CO_2_-assisted gasification. It is indirectly consistent with the fact that iron oxide-containing particles being the smallest from the series does not improve the conversion of CO_2_ significantly and the more affecting parameter is the particle nature. It is somewhat contradictory to common knowledge that the larger the particles, the smaller the fraction of atoms available on the surface of the particle. That is why the proposition that the main effect in the catalytic behaviour improvement is associated mostly with the nature of the active phase (probably metallic or oxidic nanoparticles) rather than with the particle size seems to be justified. 

The presence of maxima in the curves can be attributed to the almost-complete burning out of the samples with Co and Ni. The results of weighting of the residues after the gasification are in agreement with these results: the sample loaded with iron demonstrates a mass that is 3–5 times larger than the others. The mass of the residue of the pure sugarcane bagasse has nearly the same value as it was for the iron-loaded sample. But the iron-loaded sample residues also contain iron compounds, so it can be concluded that the completeness of the burning of carbon in the case of the sample loaded with Fe is larger than that for the pure sugar cane bagasse. Thus, the activity of the materials in CO_2_-assisted gasification decreases in the following order: Co ≈ Ni > Fe > pure.

The element composition of the samples was revealed by CHNS analyser, and the results are shown in Table 1. The initial material contains about 50 wt % of carbon. It provides a chance to estimate the integral conversions of CO_2_ and of carbon in the substrate during the overall reaction time. The calculation was performed in the range of temperature 500–850 °C, assuming the reaction CO_2_ + C = 2CO takes place exclusively. If so, the decrease in chromatographic peak of CO_2_ value corresponds to half of the amount of CO produced at the moment of the probe injection. Thus, the estimated integral of CO_2_ conversion normalised to the total area under 100% conversion in the range of 500–800 °C, providing a yield of CO on CO_2_ basis. Analogically, the estimation of CO yield can be calculated by the initial material carbon content, assuming a certain composition of nitrates in a fresh material. Here, we provide results obtained, assuming the same chemical states of metals in the sample which were used during synthesis (Figure 8).

It can be easily seen that the samples loaded with Co and Ni demonstrate relatively high yields of CO (CO_2_ basis) irrespective of the metal loading, while if the calculations were performed on a SCB carbon basis, the yields increase as metal loading increases. Iron-loaded samples show low CO yields close to that of pure lignin. Among the samples with 1 and 3 wt % metal loadings, the Co-loaded sample demonstrates higher yields than that of the Ni-loaded sample.

## 4. Discussion

The dependencies observed by SEM-EDX showed that there is no clear correlation between the uniformity of the metal distribution over the surface and the catalytic behaviour. Despite this, we can see that the metal depositions tend to be found in the natural tubes (capillaries)) of the plant. The possible reason for this effect is both the different chemical composition of the surface of the capillaries and the ‘bulk’ material.

In the catalytic tests, the metal loading does not affect the catalytic behaviour in the case of each metal in CO_2_-assisted sugar cane gasification. This may be explained as follows: there is a kind of an ‘equilibrium’ number of active species of metal oxides (or more probably metal nanoparticles), and a further increase in metal loading only leads to the formation of the same atomic sites. The observation that the conversion of CO_2_ is not affected by the metal loading provides us with a hypothesis that the process has a ‘zero order’ with respect to the metal active sites.

The possible nature of such active sites is nanoparticles in the metallic state: this is a possible reason of the poor activity of iron-loaded sugar cane bagasse: Ni and Co form more stable metal nanoparticles than iron, which is pyrophoric.

It is interesting to compare the obtained results to the previous works devoted to the process of CO_2_-assisted gasification of hydrolysis lignin. In the work [58], the activities of the samples decreased in the following order: Co > Fe > Ni, while in this work the activities were the following: Co > Ni >> Fe. The difference in these dependences may be explained by the nature of the material: hydrolytic lignin consists mostly of aromatic polymeric substances, while sugar cane contains much more functional groups in carbohydrate polymeric chains. Nevertheless, it is only a tentative hypothesis, despite the observable composition. Also, compared to our previous work devoted to lignin gasification, it can be noticed that in this work we have a much more reactive material (here we can see maxima on the CO_2_ conversion curves resulted from complete burning out of the gasified material) because of the much higher oxidation extent of both the surface and bulk. Indeed, the higher the amounts of oxygen atoms in the composition, the easier it is for the material to be gasified. By the way, the observations discussed above are correct only for the samples loaded with metals. Without such a loading, the conversion of carbon dioxide is less than that for lignin. 

The literature refers the different metal sites effects to the different diffusivity of carbon atoms through metal nanoparticles, assuming that the catalytic reaction takes place on their surface and carbon is transported through metal by diffusion [60]. The difference between the metals is explained by the different atomic radii ratios of metals (Co—1.26 Å, Ni—1.24 Å, Fe—1.32 Å) [61] and carbon (0.76–0.77 Å) atoms. It can be seen that iron atoms have a slightly larger size, while Co and Ni have nearly the same radii. On the other hand, the role of metal sites in carbon–carbon bond activation is also proposed [41].

## 5. Conclusions

The series of sugar cane bagasse-based materials with deposited metals (Fe, Co, or Ni) was prepared while varying the amounts of each metal. 

It was shown that the uniformity of oxide particles distribution on the surface decreased as soon as the metal loading increased. Simultaneously, it was shown by SEM with EDX analyses that metal oxide particles tend to be deposited inside the natural channels (plant capillaries), possibly because of the specific localization of the functional groups providing adsorption sites of metal ions. 

The dependence of the particle size distribution revealed by TEM did not demonstrate any clear effect on the catalytic performance of the samples. But the diffraction data analysis revealed the formation of metallic particles of Co and Ni at 600 °C, which seemed to be more stable than the appropriate iron particles. These observations provide us with a tentative hypothesis that the catalytic behaviour possibly results from the formation of such phases, while iron oxides are not active enough in the process of gasification. 

The catalytic tests showed that despite the similar catalytic behaviour (CO_2_ conversion up to ~80%), Co-loaded materials demonstrate a larger total yield of CO at 1 and 3 wt. % Co loading (~5% larger vs. the same Ni loading), so it can be concluded that Co is preferable for the process.

## Figures and Tables

**Figure 1 materials-16-05662-f001:**
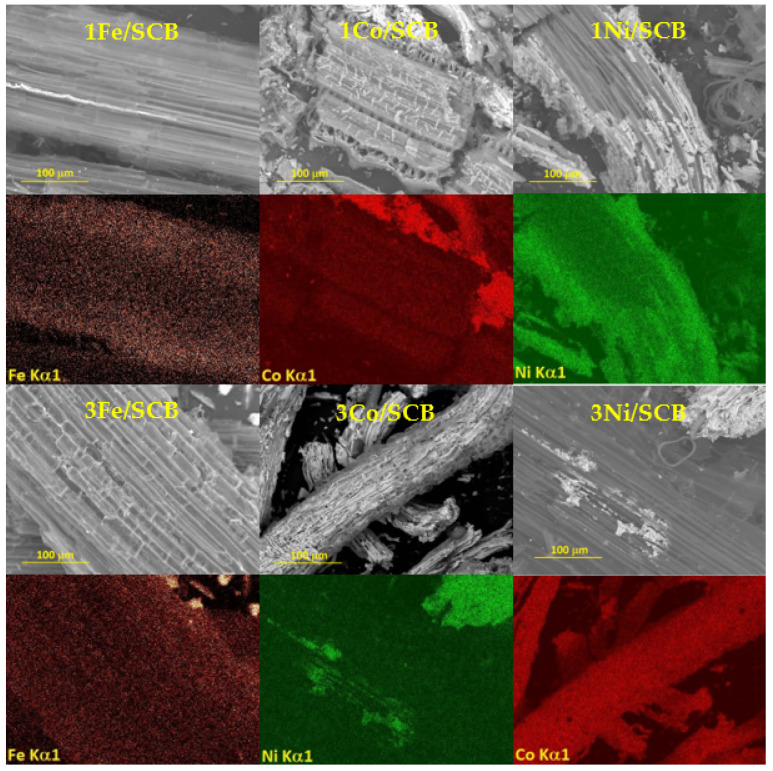
SEM images and element maps for all the catalyst after 600 °C pre-treatment.

**Figure 2 materials-16-05662-f002:**
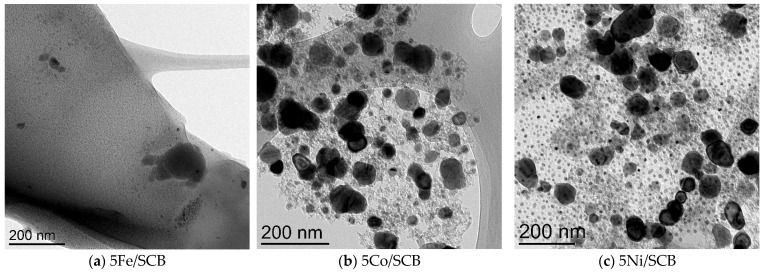
(**a**–**f**) TEM images for the samples after pre-heating at 600 °C in a CO_2_ flow at different scales; (**g**–**i**) particle size distributions for these samples (derived from measurement of about 100 particles in each case) for the samples with 5% metal mass loading 5Fe/SCB, 5Co/SCB, and 5Ni/SCB.

**Figure 3 materials-16-05662-f003:**
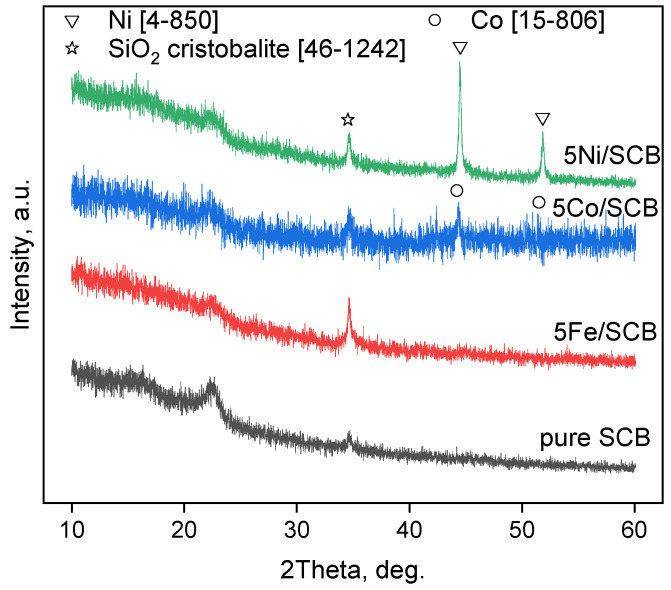
XRD patterns for the samples after pre-heating at 600 °C in a CO_2_ flow.

**Figure 4 materials-16-05662-f004:**
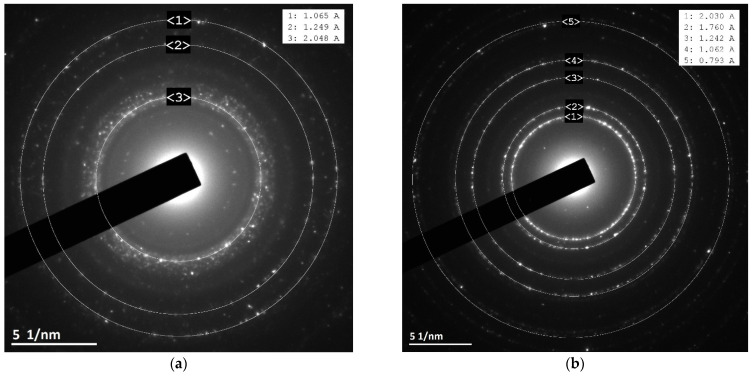
Electron diffraction patterns for the samples (**a**) 5Co/SCB and (**b**) 5Ni/SCB after pre-heating at 600 °C in CO_2_ flow. The insertions are the lists of interplanar spacings derived from the patterns.

**Figure 5 materials-16-05662-f005:**
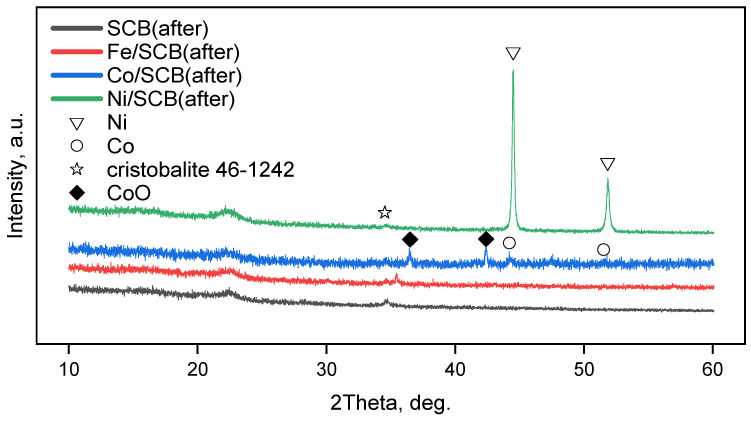
XRD patterns for the samples with 5% mass metal loading after the catalytic tests in CO_2_-assisted gasification.

**Figure 6 materials-16-05662-f006:**
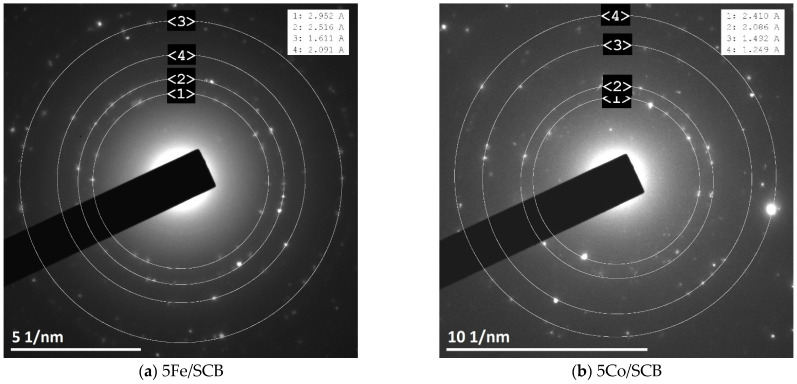
Electron diffraction patterns for the samples (**a**) 5Fe/SCB, (**b**) 5Co/SCB, (**c**) 5Ni/SCB, and (**d**) pure SCB after the catalytic tests in CO_2_-assisted gasification. The insertions are the lists of interplanar spacings derived from the patterns.

**Figure 7 materials-16-05662-f007:**
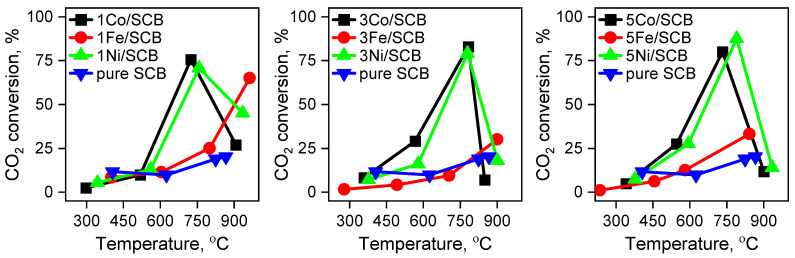
The results of the catalytic tests in the CO_2_-assisted gasification of sugar cane loaded with transition metal nitrates.

**Figure 8 materials-16-05662-f008:**
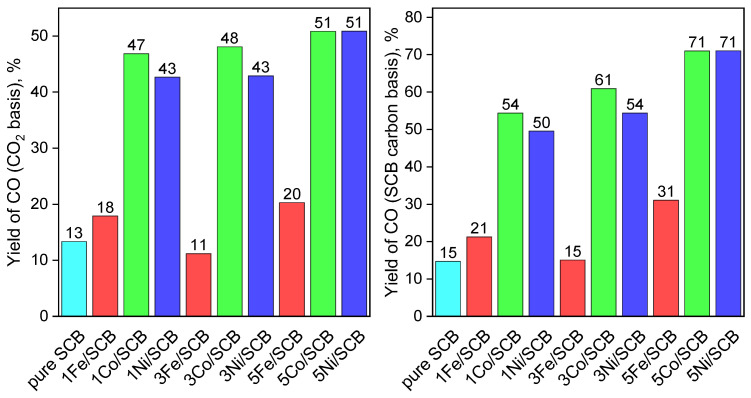
The integral yields of CO calculated on a CO_2_ basis (**left**) and on a lignin carbon basis (**right**).

**Table 1 materials-16-05662-t001:** CHNS element composition of pure SCB with standard deviations (SD).

C, wt. %	H, wt. %	N, %	S, %
46.81	6.25	0.48	<0.1

## Data Availability

Data are available upon request.

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
