# Peer review of "CO2-Assisted Sugar Cane Gasification Using Transition Metal Catalysis: An Impact of Metal Loading on the Catalytic Behavior"

_materials, 2023, doi:10.3390/ma16165662_

Round 1

Reviewer 1 Report

Manuscript ID: materials-2510764: CO2-assisted sugar cane gasification using transition metal catalysis: an impact of metal loading on the catalytic behaviour

1.      abstract should be informative, quantitative and also remove the grammatical errors?

2.      Novelty of study is missing in the introduction section?

3.      Authors are not mention numbering of heading section and subsection?

4.      In introduction citation should be improve with recent literatures?

5.      Citation of result discussion section is very weak, so improved through recent studies in field in all sections? Without these improvement in the manuscript not be like publish.

6.      Conclusion of manuscript will be concise and with quantitative results and also merged all points?

7.      Referencing should be as per journal guidelines.

Manuscript ID: materials-2510764: CO2-assisted sugar cane gasification using transition metal catalysis: an impact of metal loading on the catalytic behaviour

1.      abstract should be informative, quantitative and also remove the grammatical errors?

2.      Novelty of study is missing in the introduction section?

3.      Authors are not mention numbering of heading section and subsection?

4.      In introduction citation should be improve with recent literatures?

5.      Citation of result discussion section is very weak, so improved through recent studies in field in all sections? Without these improvement in the manuscript not be like publish.

6.      Conclusion of manuscript will be concise and with quantitative results and also merged all points?

7.      Referencing should be as per journal guidelines.

Author Response

We sincerely thank our referees for their valuable comments and suggestions on improving our manuscript. All the comments are listed below with our responses.

Reviever 1

1) abstract should be informative, quantitative and also remove the grammatical errors?

The abstract was re-written according your recommendations and now it looks like below.

Abstract: To meet the increasing needs in fuels, especially non-fossil, the production of “bio-oil” is proposed and a lot of efforts are done to find the effective ways to transform bio-wastes into the valuable substances to obtain the fuels and simultaneously reduce the carbon wastes including CO2. This work is devoted to the gasification of sugar cane bagasse to produce CO in the process assisted by CO2. The metals were varied (Fe, Co, or Ni) along with their amounts in order to find the optimal catalyst composition. The materials were investigated by SEM, EDX, TEM, XRD and electron diffraction and tested in the process of CO2-assisted gasification. The catalysts based on Co and Ni demonstrate the best activity among the investigated systems: the conversion of CO2 reached 88 % at ~800 °C (vs 20% for the pure sugarcane bagasse). These samples contain metallic Co or Ni, while Fe is in oxide form.

2) Novelty of study is missing in the introduction section?

The novelty of this study was highlighted, and now it looks like below.

The main novelty of this work is related to revealing the correlation between the nature of the metal deposited on sugarcane bagasse, the resulting structural properties, and the catalytic activity of the materials.

3) Authors are not mention numbering of heading section and subsection?

The mistakes in numbering of the subsections were corrected.

4) In introduction citation should be improve with recent literatures?s

The following recent studies were cited additionally to improve this section.

  • Rafiee, A.; Rajab Khalilpour, K.; Milani, D.; Panahi, M. Trends in CO2 Conversion and Utilization: A Review from Process Systems Perspective. Environ. Chem. Eng. 2018, 6 (5), 5771–5794. https://doi.org/10.1016/j.jece.2018.08.065.
  • Wang, X.; Chen, Q.; Zhu, H.; Chen, X.; Yu, G. In-Situ Study on Structure Evolution and Gasification Reactivity of Biomass Char with K and Ca Catalysts at Carbon Dioxide Atmosphere. Carbon Resour. Convers. 2023, 6 (1), 27–33. https://doi.org/10.1016/j.crcon.2022.10.002.
  • Zhang, S.; Wang, J.; Ye, L.; Li, S.; Su, Y.; Zhang, H. Investigation into Biochar Supported Fe-Mo Carbides Catalysts for Efficient Biomass Gasification Tar Cracking. Eng. J. 2023, 454, 140072. https://doi.org/10.1016/j.cej.2022.140072.
  • Faust, R.; Valizadeh, A.; Qiu, R.; Tormachen, A.; Maric, J.; Vilches, T. B.; Skoglund, N.; Seemann, M.; Halvarsson, M.; Öhman, M.; Knutsson, P. Role of Surface Morphology on Bed Material Activation during Indirect Gasification of Wood. Fuel 2023, 333. https://doi.org/10.1016/j.fuel.2022.126387.
  • Kim, J. Y.; Kim, D.; Li, Z. J.; Dariva, C.; Cao, Y.; Ellis, N. Predicting and Optimizing Syngas Production from Fluidized Bed Biomass Gasifiers: A Machine Learning Approach. Energy 2023, 263, 125900. https://doi.org/10.1016/j.energy.2022.125900.
  • Mei, Y.; Zhang, Q.; Wang, Z.; Gao, S.; Fang, Y. Novel Re-Utilization of High-Temperature Catalytic Gasification Ash with Sodium Recovery, Aluminum Extraction, Aragonite and Mesoporous SiO2 Fuel 2023, 331, 125727. https://doi.org/10.1016/j.fuel.2022.125727.
  • Ma, Y.; Zha, Z.; Huang, C.; Ge, Z.; Zeng, M.; Zhang, H. Gasification Characteristics and Synergistic Effects of Typical Organic Solid Wastes under CO2/Steam Atmospheres. Waste Manag. 2023, 168 (December 2022), 35–44. https://doi.org/10.1016/j.wasman.2023.05.040.
  • Varga, G.; Sápi, A.; Varga, T.; Baán, K.; Szenti, I.; Halasi, G.; Mucsi, R.; Óvári, L.; Kiss, J.; Fogarassy, Z.; Pécz, B.; Kukovecz, Á.; Kónya, Z. Ambient Pressure CO2 Hydrogenation over a Cobalt/Manganese-Oxide Nanostructured Interface: A Combined in Situ and Ex Situ Study. Catal. 2020, 386, 70–80. https://doi.org/10.1016/j.jcat.2020.03.028
  • Tolkachev, N. N.; Koklin, A. E.; Laptinskaya, T. V.; Lunin, V. V.; Bogdan, V. I. Influence of Heat Treatment on the Size of Sodium Lignosulfonate Particles in Water—Ethanol Media. Chem. Bull. 2019, 68 (8), 1613–1620. https://doi.org/10.1007/s11172-019-2600-6.
  • Kulikov, L. A.; Bazhenova, M. A.; Makeeva, D. A.; Terenina, M. V; Maximov, A. L.; Karakhanov, E. A. Hydrogenation of Lignin Bio-Oil Components over Catalysts Based on Porous Aromatic Frameworks. Chem. 2022, 62 (9), 1096–1106. https://doi.org/10.1134/S096554412209002X
  • Tolkachev, N. N.; Koklin, A. E.; Laptinskaya, T. V.; Lunin, V. V.; Bogdan, V. I. Influence of Heat Treatment on the Size of Sodium Lignosulfonate Particles in Water—Ethanol Media. Chem. Bull. 2019, 68 (8), 1613–1620. https://doi.org/10.1007/s11172-019-2600-6.
  • Koklin, A. E.; Bobrova, N. A.; Bogdan, T. V.; Mishanin, I. I.; Bogdan, V. I. Conversion of Phenol and Lignin as Components of Renewable Raw Materials on Pt and Ru-Supported Catalysts. Molecules 2022, 27 (5). https://doi.org/10.3390/molecules27051494.
  • Bogdan, T. V; Bobrova, N. A.; Koklin, A. E.; Mishanin, I. I.; Odintsova, E. G.; Antipova, M. L.; Petrenko, V. E.; Bogdan, V. I. Structure of Aqueous Solutions of Lignin Treated by Sub- and Supercritical Water: Experiment and Simulation. Mol. Liq. 2023, 383, 122030. https://doi.org/https://doi.org/10.1016/j.molliq.2023.122030.
  • Singh, O.; Sharma, T.; Ghosh, I.; Dasgupta, D.; Vempatapu, B. P.; Hazra, S.; Kustov, A. L.; Sarkar, B.; Ghosh, D. Converting Lignocellulosic Pentosan-Derived Yeast Single Cell Oil into Aromatics: Biomass to Bio-BTX. ACS Sustain. Chem. Eng. 2019, 7 (15), 13437–13445. https://doi.org/10.1021/acssuschemeng.9b02851.
  • Naranov, E.; Sadovnikov, A.; Arapova, O.; Kuchinskaya, T.; Usoltsev, O.; Bugaev, A.; Janssens, K.; De Vos, D.; Maximov, A. The In-Situ Formation of Supported Hydrous Ruthenium Oxide in Aqueous Phase during HDO of Lignin-Derived Fractions. Catal. B Environ. 2023, 334, 122861. https://doi.org/https://doi.org/10.1016/j.apcatb.2023.122861.
  • Zharova, P.; Arapova, O. V.; Konstantinov, G. I.; Chistyakov, A. V.; Tsodikov, M. V. Kraft Lignin Conversion into Energy Carriers under the Action of Electromagnetic Radiation. Chem. 2019. https://doi.org/10.1155/2019/6480354.
  • Netrusov, A. I.; Teplyakov, V. V; Tsodikov, M. V; Chistjakov, A. V; Zharova, P. A.; Shalygin, M. G. Laboratory Scale Production of Hydrocarbon Motor Fuel Components from Lignocellulose: Combination of New Developments of Membrane Science and Catalysis. Biomass and Bioenergy 2020, 135, 105506. https://doi.org/https://doi.org/10.1016/j.biombioe.2020.105506.
  • Tsodikov, M. V.; Nikolaev, S. A.; Chistyakov, A. V.; Bukhtenko, O. V.; Fomkin, A. A. Formation of Adsorbents from Fe-Containing Processing Residues of Lignin. Microporous Mesoporous Mater. 2020, 298. https://doi.org/10.1016/j.micromeso.2020.110089.
  • Tsodikov, M. D.; Ellert, O. G.; Arapova, O. V.; Nikolaev, S. A.; Chistyakov, A. V.; Maksimov, Y. V. Benefit of Fe-Containing Catalytic Systems for Dry Reforming of Lignin to Syngas under Microwave Radiation. Eng. Trans. 2018, 65, 367–372. https://doi.org/10.3303/CET1865062.

5) Citation of result discussion section is very weak, so improved through recent studies in field in all sections? Without these improvement in the manuscript not be like publish.

The section was updated with the citations of the recent works in the field.

6) Conclusion of manuscript will be concise and with quantitative results and also merged all points?

Abstract section was updated and now it looks like presented below.

Conclusions

The series of sugar cane bagasse-based materials with deposited metals (Fe, Co, or Ni) was prepared while varying the amounts of each metal.

It was shown that the uniformity of oxide particles distribution on the surface decreases as soon as the metal loading increases. Simultaneously it was shown by SEM with EDX analyses that metal oxide particles tend to be deposited inside the natural channels (plant capillaries) possibly because of specific localization of the functional groups providing adsorption sites of metal ions.

The dependence of the particle size distribution revealed by TEM did not demonstrate any clear effect on the catalytic performance of the samples. But the diffraction data analysis revealed the formation of metallic particles of Co or Ni already at 600 °C, which seem to be more stable than the appropriate iron particles. These observations provide us with a tentative hypothesis that the catalytic behaviour possibly results from the formation of such phases while iron oxides are not active enough in the process of gasification.

The catalytic tests showed that despite the similar catalytic behaviour (CO2 conversion up to ~80%), Co-loaded materials demonstrate larger total yield of CO at 1 and 3 wt % Co loading (~5% larger vs the same Ni loading) so it can be concluded that Co is preferable for the process.

  1. Referencing should be as per journal guidelines.

References were formatted accordingly the guidelines of the journal.

Reviewer 2 Report

This manuscript presents a systematic study on the design and development of a CO2-assisted transition metal nanoparticle catalyzed gasification of sugar cane bagasse. Both cobalt- and iron- and nickel-catalyzed reactions were investigated and a clear trend of their catalytic activities was found.

While the Co- and Ni-based reactions were found to be almost identical in terms of efficiency, both the iron-based and transition metal-free reactions performed worse compared to the previous ones.

The work presented is well designed and has clearly shown the strengths of the process developed.

There is only one weakness of this manuscript. In order to clearly understand the potential and role of metal centers, a comparative table should be added to the main article, including the catalytic performance of already well-studied transition metal/transition metal ion-based systems for CO2 hydrogenation with emphasis on the production of CO. On this basis, it would be explained why Co- and Ni-containing systems are able to effectively produce syngas in the presented process.

To this end, however, the introduction must also be upgraded (e.g. using the follow publication: G. Varga et al., “Ambient pressure CO2 hydrogenation over a cobalt/manganese-oxide nanostructured interface: A combined in situ and ex situ study” J. Catal. 386 (2020) 70–80).

Considering the above-mentioned points, I suggest this manuscript for publication after major revision.

Author Response

We sincerely thank our referees for their valuable comments and suggestions on improving our manuscript. All the comments are listed below with our responses.

Reviewer 2

There is only one weakness of this manuscript. In order to clearly understand the potential and role of metal centers, a comparative table should be added to the main article, including the catalytic performance of already well-studied transition metal/transition metal ion-based systems for CO2 hydrogenation with emphasis on the production of CO. On this basis, it would be explained why Co- and Ni-containing systems are able to effectively produce syngas in the presented process.

This study is mostly focused on the gasification of carbon material to produce CO, but the main reaction here is C(s) + CO2(g) = 2CO(g). CO2 hydrogenation into methane takes place inly at the temperatures below ~500 °C while the process of gasification (i.e., carbon removal) occurs at higher temperatures. Methanation plays a little role in the overall process because its contribution into sum conversion is relatively small. That is why the comparison between our results and the results of the researches of CO2 hydrogenation does not seem to be correct here. The metal sites effects are the result from the different diffusivity of the atoms through the carbon matrix according to [LS Lobo, S.A.C. Carabineiro, Fuel, V. 183, 2016, Pp. 457–469, https://doi.org/10.1016/j.fuel.2016.06.115].

So, the manuscript was updated with the following passage.

Literature refers the different metal sites effects to the different diffusivity of carbon atoms through metal nanoparticles assuming that the catalytic reaction takes plays on their surface and carbon is transported through metal by diffusion [49]. The difference between the metals is explained by the different atomic radii ratios of metals (Co — 1.26 Å, Ni — 1.24 Å, Fe — 1.32 Å) [50] and carbon (0.76–0.77 Å) atoms. It can be seen that iron atoms have slightly larger size while Co and Ni have nearly the same radii.

To this end, however, the introduction must also be upgraded (e.g. using the follow publication: G. Varga et al., “Ambient pressure CO2 hydrogenation over a cobalt/manganese-oxide nanostructured interface: A combined in situ and ex situ study” J. Catal. 386 (2020) 70–80).

Of course, the Introduction section was updated with this interesting article.

Considering the above-mentioned points, I suggest this manuscript for publication after major revision.

Round 2

Reviewer 2 Report

The authors have been responsive to my suggestions and concerns and have done a good job of improving the relevant sections. In the light of this, I suggest this manuscript for publication in an MDPI journal "Materials" in this present form.